# Genetic Spectrum of Syndromic and Non-Syndromic Hearing Loss in Pakistani Families

**DOI:** 10.3390/genes11111329

**Published:** 2020-11-11

**Authors:** Julia Doll, Barbara Vona, Linda Schnapp, Franz Rüschendorf, Imran Khan, Saadullah Khan, Noor Muhammad, Sher Alam Khan, Hamed Nawaz, Ajmal Khan, Naseer Ahmad, Susanne M. Kolb, Laura Kühlewein, Jonathan D. J. Labonne, Lawrence C. Layman, Michaela A. H. Hofrichter, Tabea Röder, Marcus Dittrich, Tobias Müller, Tyler D. Graves, Il-Keun Kong, Indrajit Nanda, Hyung-Goo Kim, Thomas Haaf

**Affiliations:** 1Institute of Human Genetics, Julius Maximilians University Würzburg, 97074 Würzburg, Germany; julia.doll@uni-wuerzburg.de (J.D.); schnapp-linda@web.de (L.S.); susi_kolb@yahoo.de (S.M.K.); michaela.hofrichter@uni-wuerzburg.de (M.A.H.H.); tabea.roeder@uni-wuerzburg.de (T.R.); marcus.dittrich@biozentrum.uni-wuerzburg.de (M.D.); nanda@biozentrum.uni-wuerzburg.de (I.N.); thomas.haaf@uni-wuerzburg.de (T.H.); 2Tübingen Hearing Research Centre, Department of Otolaryngology-Head and Neck Surgery, Eberhard Karls University, 72076 Tübingen, Germany; 3Max Delbrück Center for Molecular Medicine in the Helmholtz Association, 13125 Berlin, Germany; fruesch@mdc-berlin.de; 4Department of Chemistry, Bacha Khan University Charsadda, Khyber Pakhtunkhwa 24420, Pakistan; imrangnu@gmail.com; 5Department of Biotechnology and Genetic Engineering, Kohat University of Science and Technology, Kohat, Khyber Pakhtunkhwa 24420, Pakistan; saadkhanwazir@gmail.com (S.K.); noormwazir@yahoo.com (N.M.); sakmarwat79@gmail.com (S.A.K.); hamedwazir@gmail.com (H.N.); 6Department of Biotechnology, Bacha Khan University Charsadda, Khyber Pakhtunkhwa 24420, Pakistan; ajmalkhanbbt@gmail.com; 7District Eye Specialist, Police and Services Hospital Peshawar, Khyber Pakhtunkkhwa 24420, Pakistan; na_safi1982@yahoo.com; 8Department of Ophthalmology, Eberhard Karls University, 72076 Tübingen, Germany; laura.kuehlewein@med.uni-tuebingen.de; 9Section of Reproductive Endocrinology, Infertility & Genetics, Department of Obstetrics and Gynecology, Medical College of Georgia, Augusta University, Augusta, GA 30912, USA; molgenetics_and_epigenetics@hotmail.com (J.D.J.L.); lalayman@augusta.edu (L.C.L.); tylergraves14@hotmail.com (T.D.G.); 10Department of Neuroscience and Regenerative Medicine, Medical College of Georgia, Augusta University, Augusta, GA 30912, USA; 11Department of Bioinformatics, Julius Maximilians University Würzburg, 97074 Würzburg, Germany; tobias.mueller@biozentrum.uni-wuerzburg.de; 12Department of Animal Sciences, Division of Applied Life Science (BK21 Four), Gyeongsang National University, Jinju 52828, Korea; ikong7900@gmail.com; 13Neurological Disorders Research Center, Qatar Biomedical Research Institute, Hamad Bin Khalifa University, 34110 Doha, Qatar

**Keywords:** genetic diagnosis, consanguinity, genome-wide linkage analysis, hearing loss, Pakistan, exome sequencing

## Abstract

The current molecular genetic diagnostic rates for hereditary hearing loss (HL) vary considerably according to the population background. Pakistan and other countries with high rates of consanguineous marriages have served as a unique resource for studying rare and novel forms of recessive HL. A combined exome sequencing, bioinformatics analysis, and gene mapping approach for 21 consanguineous Pakistani families revealed 13 pathogenic or likely pathogenic variants in the genes *GJB2*, *MYO7A*, *FGF3*, *CDC14A*, *SLITRK6*, *CDH23*, and *MYO15A*, with an overall resolve rate of 61.9%. *GJB2* and *MYO7A* were the most frequently involved genes in this cohort. All the identified variants were either homozygous or compound heterozygous, with two of them not previously described in the literature (15.4%). Overall, seven missense variants (53.8%), three nonsense variants (23.1%), two frameshift variants (15.4%), and one splice-site variant (7.7%) were observed. Syndromic HL was identified in five (23.8%) of the 21 families studied. This study reflects the extreme genetic heterogeneity observed in HL and expands the spectrum of variants in deafness-associated genes.

## 1. Introduction

In parts of the world where consanguinity is prevalent, it is not uncommon to see a high prevalence of genetic diseases. The consanguineous marriage rates in Pakistan are among the highest worldwide [1]. Approximately 60% of marriages in Pakistan are consanguineous, with roughly 80% of these marriages being between first cousins [2]. The prevalence of autosomal recessive diseases associated with a monogenic background, such as profound hearing loss (HL), is high in countries where consanguineous marriages are common [3]. 

Hereditary HL is one of the most prevalent sensory disorders that affects 1 to 2 per 1000 live births worldwide. Genetic factors are responsible for over half of all HL [4]. Studies describing genetic variants in Pakistani families with HL show evidence of the extreme clinical and genetic heterogeneity of this sensory disease and support the importance of investigating and characterizing families from this region of the world [5,6,7]. Over 120 genes have been identified as causing non-syndromic hearing loss (NSHL), which comprises approximately 70% of all forms of hereditary HL (http://hereditaryhearingloss.org). Autosomal recessive HL (ARHL) is the most commonly observed inheritance pattern. There are presently over 600 syndromic forms of deafness [8], which appear in approximately 30% of patients with genetic HL. Many of these deafness syndromes mimic non-syndromic deafness at onset [9]. Hearing impairment profoundly complicates speech and language development in prelingual children and can negatively impact education and employment prospects [10].

Exome sequencing (ES) allows for the parallel sequencing of all coding regions of the human genome and has accelerated the process of identifying causally associated variants in patients with HL [11]. In 13 consanguineous families with diverse forms of HL, we identified 13 variants in 7 HL-associated genes using ES and gene mapping approaches in 21 Pakistani families. Two of the 13 variants were not previously described in the literature. The present study underscores the importance of genetically characterizing consanguineous families with HL to expand the spectrum of clinically relevant variants in genetically diverse populations, thus improving our understanding of the alleles involved in ARHL and enhancing genetic counseling.

## 2. Materials and Methods 

### 2.1. Clinical Evaluation

This study was approved by the Ethics Committees of Augusta University (624456-4), Kohat University of Science and Technology (16–25), and the University of Würzburg (46/15). Fully informed written consent was obtained prior to initiating our study. Informed written consent from minors was provided from parent(s) or legal guardians. We recruited 21 consanguineous Pakistani families with congenital, bilateral, and severe-to-profound HL. The affected individuals in family 6 and 7 were audiologically tested by pure-tone audiometry, conforming with the established guidelines described by Mazzoli et al. [12]. Hearing thresholds were measured at 0.25, 0.5, 1, 2, 4, 6, and 8 kHz. HL was self-reported in all other families but clearly noted as severe-to-profound. Ophthalmic examinations of families 4, 5, 6, 7, and 8 were performed. 

### 2.2. Autozygosity Mapping and Linkage Analysis

#### 2.2.1. Genotyping and Quality Control

The Illumina Infinium HumanCore-24 v1.0 Bead Chip array (Illumina, Inc., San Diego, CA, USA) was used for genotyping. From the 306,670 markers on the array, we filtered out indels, MT- and Y-chromosomal SNPs, and variations without physical positions, resulting in 259,460 biallelic SNPs for quality control (QC) and linkage analysis. Data conversion to linkage format files and QC was managed with ALOHOMORA software [13]. The sex of individuals was estimated by counting heterozygous genotypes on the X-chromosome and compared to the upraised pedigree data. The relationships between family members were verified with the program GRR [14]. PedCheck [15] was used to detect Mendelian errors (ME) and SNPs with ME were removed from the data set. Unlikely genotypes—e.g., double recombinants—were identified with Merlin [16] and deleted in the individuals.

#### 2.2.2. Linkage Analysis

Linkage analysis was performed with Merlin using an autosomal recessive mode of inheritance and complete penetrance. We assumed 0.001 as the mutant allele frequency. We executed Merlin twice, once with a full marker set of around 258,000 SNPs after QC. This calculation was used to obtain the best positions for recombination events. The second analysis was conducted with a reduced marker set (~119,000 SNPs), where a minimal distance of 10,000 bases between markers was used. This calculation, where the linkage disequilibrium (LD) between markers is reduced, identifies linkage peaks which were inflated by markers in LD. We removed the linkage regions where the LOD score broke down more than 0.3 in the LD-reduced analysis. In summary, we selected regions where the LOD score reached the maximal LOD score of a family and where the LOD score was stable in the less dense, LD-reduced marker set. Under the given inheritance model (recessive) and the pedigree structure with a consanguinity loop, this linkage analysis is called autozygosity mapping. 

### 2.3. Exome Sequencing

Genomic DNA (gDNA) was extracted from peripheral blood lymphocytes using a standard phenol/chloroform [17] and ethanol precipitation [18]. A total of 50 ng of gDNA from the proband from each family was subjected to ES with the Nextera Rapid Capture Exome or the TruSeq Exome Enrichment kits (Illumina, Inc., San Diego, CA, USA) according to the manufacturer’s protocol. An additional family member (IV.1) of family 5 was exome sequenced due to the presence of two distinct phenotypes in the family—namely, HL and a suspected bone disorder. A 2 × 76 bp paired-end read sequencing was performed using a v2 high-output reagent kit with the NextSeq500 sequencer (Illumina, Inc., San Diego, CA, USA). Raw bcl sequencing files were converted with the bcl2fastq software (Illumina, Inc., San Diego, CA, USA) and the data were aligned to the human reference genome GRCh37 [19] (hg19).

### 2.4. Variant Analysis and Prioritization

Single-nucleotide variants (SNVs) and small indels (<15 bp) were analyzed with the GensearchNGS software (PhenoSystems SA, Wallonia, Belgium). Analysis was supported using an established in-house bioinformatics pipeline based on the GATK toolkit including Burrows-Wheeler (BWA)-based read alignment, base quality score recalibration, indel realignment, duplicate removal, and SNP and indel discovery, with subsequent score recalibration according to the GATK Best Practice recommendations [19,20,21]. Variant filtering and prioritization were performed using a conservative minor allele frequency <0.01 based on population databases and an alternate allele frequency present at >20% referring to reads. Additional variants not removed by minor allele frequency filtering were subjected to an in-house allele count filter (*n* = 300) that removed variants appearing >2%, as these are too common in our exome dataset to enter manual analysis. Variant prioritization included the tools PolyPhen-2 (PP) [22], SIFT [23], MutationTaster (MT) [24], fathmm [25], LRT [26], and GERP [27]. The Deafness Variation Database [28] was also integrated into our pipeline to permit the quick assessment of variants in known deafness genes. Frequency-based filtering was performed according to a population-specific manner that includes The Greater Middle East Variome Project [29] and gnomAD [30] to account for varying allele frequencies across ethnicities. CNV analysis was performed for 19 out of 21 families using the eXome Hidden MarkovModel (XHMM, version 1.0) approach [31].

### 2.5. Variant Validation and Segregation Testing

The candidate variants remaining after filtering were amplified by PCR using primers designed from the Primer3 software [32]. The primer sequences are shown in Appendix A. PCR products were bidirectionally sequenced with an ABI 3130xl 16-capillary sequencer (Life Technologies, Carlsbad, CA, USA). Sequence reactions were completed with 5× sequencing buffer and big dye terminator (Applied Biosystems, Waltham, MA, USA). DNA sequence analysis was performed using the Gensearch software (Phenosystems SA, Wallonia, Belgium).

## 3. Results

### 3.1. Summary of Affected Genes and Genetic Context

Using ES and bioinformatics analysis, 13 different variants in seven HL-associated genes were identified, including two that have not been previously described in the literature (15.4%). In aggregate, these variants are likely causally associated in 13 out of 21 (61.9%) consanguineous Pakistani families. Pathogenic and likely pathogenic variants were identified in the genes *GJB2*, *MYO7A*, *FGF3*, *CDC14A*, *SLITRK6*, *CDH23*, and *MYO15A* (Table 1). *GJB2* and *MYO7A* were implicated in the genetic diagnosis of almost half of all cases (46.2%). Among the different variant types observed, seven were missense (53.8%), three were nonsense (23.1%), two were frameshift (15.4%), and one was a splice-site variant (7.7%). All the variants were either homozygous or compound heterozygous, showed an autosomal recessive inheritance pattern, and were validated by segregation testing (Figure 1, Appendix A).

### 3.2. Clinical Features and Genetic Spectrum of Patients with Syndromic HL

#### *FGF3*, *MYO7A* and *SLITRK6*

All the affected individuals were clinically diagnosed with congenital, bilateral HL and have a consanguineous background. Five of 21 families (23.8%) revealed a syndromic form of HL. Two of the 21 families were clinically diagnosed with Usher syndrome, one of the most common forms of syndromic HL [8].

The affected individuals who were available for testing in families 1 and 2 reported severe HL and cupped ears (Figure 2A, Table 2) (IV.1, IV.2, IV.3, family 1; III.5, III.6, III.10, IV.2, IV.5, family 2). A homozygous missense c.166C>T, p.(Leu56Phe) variant was identified in *FGF3* (NM_005247.2; Appendix A) in families 1 and 2 that co-segregated with HL in both families (Figure 1). The variant is predicted to be disease causing (MT, PP, SIFT), involves the substitution of a conserved amino acid (aa), and was not previously published in the literature. The variant is classified as likely pathogenic according to the ClinGen HL working group expert specification [41]. Homozygous variants in *FGF3* have been associated with deafness, accompanied by inner ear agenesis, microtia, and microdontia [42].

The proband (IV.1) in family 4 and his affected siblings (IV.2, IV.3, IV.5) were diagnosed with Usher syndrome and revealed a homozygous pathogenic missense c.470G>A, p.(Ser157Asn) variant [34] in *MYO7A*. Ophthalmological examination of the proband IV.3 revealed high myopia, cataract, and retinitis pigmentosa in both eyes. Visual acuity was reduced to 5/60 (Snellen equivalent, 20/250), with corrective lenses of −12.00/0/0° in both eyes. Ophthalmological examination of the proband IV.5 revealed high myopia, cataract, and retinitis pigmentosa in both eyes. Visual acuity was reduced to light perception in the right eye, and 1/60 (Snellen equivalent, worse than 20/1000) in the left eye. Thus, the patient was legally blind. Cataract was more pronounced in the right eye. Ultrasonographic findings of the right eye were within normal limits. The aa substitution weakens the 5′ donor splice-site predicted by several in silico prediction tools. A previously published minigene assay that was conducted using nasal epithelial cells proved the skipping of exon 5 in the mutant transcript, which likely results in a truncated protein [43] (Figure 1). Two individuals in family 5 (III.4, IV.1) suffer from a bone disorder but have normal hearing in contrast to the affected family members (IV.2, IV.3), who show a distinct Usher syndrome phenotype (Table 2). The ophthalmological examination of proband IV.2 revealed hyperopia in both eyes (+9/0/0° in the right eye, and +11/0/0° in the left eye) and lenticular opacity (cataract) in both eyes. Retinitis pigmentosa was confirmed with indirect ophthalmoscopy with a 20 Diopter power lens. We identified a homozygous pathogenic missense c.3502C>T, p.(Arg1168Trp) variant [35] in the gene *MYO7A* in affected individuals (IV.2, IV.3) that segregated in family 5. An ES analysis of III.3 and IV.1 did not uncover any variants in genes associated with bone disorders or neuropathies. Both variants that have been identified in family 4 and family 5 are known to cause Usher syndrome type 1 [34,35].

The affected individuals in family 7 (IV.1, IV.2) reported severe-to-profound HL (Table 2, Figure 2B). Ophthalmological examination in IV.1 revealed compound myopic astigmatism and grossly cupped discs diagnosed as glaucoma in both eyes. The macula appeared normal in both eyes. Visual acuity was reduced to 0.5 logMAR in both eyes (Snellen equivalent: 20/63), with corrective lenses of −5.50/−2.50/90° in the right eye and −6.50/−1.00/60° in the left eye. Intraocular pressure was 17 mmHg in both eyes. Additionally, Duane retraction syndrome (a congenital neuromuscular dysfunction of the eye movement caused by a failure of the sixth cranial nerve) was diagnosed in the left eye. Other data (e.g., axial length, status of lens, macula/retina, visual field, treatment of glaucoma) were not available. Ophthalmological examination in IV.2 revealed compound myopic astigmatism and myopic alterations with macular degeneration in both eyes. Visual acuity was reduced to count fingers with corrective lenses of −5.00/−2.50/90° in the right eye and −4.00/−3.00/90° in the left eye. Intraocular pressure was 15 mmHg in both eyes. Thus, the patient was legally blind. Other data (e.g., axial length, status of lens, macula/retina, visual field, treatment of glaucoma) were not available. Family 7 revealed a segregating novel homozygous nonsense variant c.120_121insT, p.(Asp41*) in the gene *SLITRK6* (NM_032229.2; Appendix A) that was present in both affected family members (IV.1, IV.2) (Figure 1) and was absent in population databases. The variant is classified as likely pathogenic according to the ClinGen HL working group expert specification [41]. Homozygous variants in this gene are known to cause sensorineural deafness and high myopia in humans [44].

Both previously unreported variants have been submitted to the Leiden Open Variation Database (LOVD) v3.0 under the accession IDs 00307903 (*FGF3* c.166C>T, p.(Leu56Phe)) and 00307904 (*SLITRK6* c.120_121insT, p.(Asp41*)).

### 3.3. Identification of Causative Variants in Patients with NSHL

#### 3.3.1. *GJB2*

Variants in the gene *GJB2* (NM_004004.5; DFNB1A, DFNA3A) were identified in three families (3, 10, 11). The proband (IV.1) and his affected sibling (IV.2) in family 3 revealed a common homozygous pathogenic nonsense variant c.231G>A, p.(Trp77*) [33] that was consistent with the familial segregation analysis (Figure 1). The probands in families 10 (IV.3) and 11 (IV.3) revealed a prevalent homozygous pathogenic frameshift variant c.35delG, p.(Gly12Valfs*2) [40] thatsegregated in both families (Figure 1). In this study, variants in *GJB2* accounted for 23.1% of the 13 resolved families and comprised 14.3% (3 out of 21) of the total diagnostic yield.

#### 3.3.2. *MYO7A*

Family 8 revealed three different heterozygous *MYO7A* variants: c.1258A>T, p.(Lys420*) [37], c.1849T>C, p.(Ser617Pro) [38], and c.4505A>G, p.(Asp1502Gly) [6]. All three heterozygous variants were identified exclusively in the affected individuals (IV.1, IV.2). The unaffected mother (III.4) revealed only two of the variants (c.1258A>T, p.(Lys420*); c.4505A>G, p.(Asp1502Gly)) and the unaffected paternal grandfather (II.1) was confirmed with the third variant c.1849T>C, p.(Ser617Pro), confirming a compound heterozygosity of p.(Lys420*) and p.(Ser617Pro) (Figure 1). Vision was normal in all the affected individuals, and Usher syndrome was not confirmed. Ophthalmological examination in proband IV.1 revealed no significant ocular problems in either eye. Visual acuity was 6/6 (Snellen equivalent, 20/20) in both eyes. Photographs of the central retina showed the optic disc, macula, and vessels within normal limits in both eyes.

In aggregate, *MYO7A* (NM_000260.3; DFNA11, DFNB2, USH1B) was affected in three families, accounting for 23.1% of the overall diagnostic yield.

#### 3.3.3. *CDC14A*, *CDH23* and *MYO15A*

Family 6 reported severe-to-profound HL and suffers from compound myopic astigmatism (IV.1, IV.2). A frameshift variant (c.1041dup, p.(Ser348Glnfs*2)) was identified in the deafness gene *CDC14A* (NM_033312.2, DFNB32) in affected family members of family 6 (IV.1, IV.2) and co-segregated with HL in this family (Figure 1). Both affected individuals are unmarried and have no children. This variant was described as disease causing supported by functional RT-qPCR validation [36]. 

We identified two families with variants in the gene *CDH23*. The proband of family 9 (IV.1) revealed a homozygous pathogenic *CDH23* (NM_022124.5, DFNB12) missense variant c.2968G>A, p.(Asp990Asn) [39] that was validated via Sanger sequencing and was also present in three affected siblings (IV.2, IV.3, IV.4) (Figure 1). The affected proband in family 13 showed a segregating homozygous pathogenic missense variant c.4688T>C, p.(Leu1563Pro) [45] in the gene *CDH23* (Figure 1).

Furthermore, we identified a homozygous splice-site variant c.9518-2A>G in *MYO15A* (NM_016239.3, DFNB3) in the proband (IV.1) and affected siblings (IV.2, IV.4) in family 12 (Figure 1). This variant likely mediates the loss of the canonical splice acceptor site predicted by several in silico prediction tools and was previously reported in another Pakistani family with NSHL [5].

### 3.4. Autosomal Recessive HL Loci

Genome-wide genotyping and autozygosity mapping that were performed for 13 Pakistani families revealed loci for autosomal recessive HL (hg19) (Table 3).

The homozygous c.166C>T, p.(Leu56Phe) variant in *FGF3* was identified and supported by linkage intervals spanning 33.5 Mb and 6.1 Mb for families 1 and 2 (Table 3), respectively. The 19.4 Mb interval in family 4 (Table 3) included the homozygous c.470G>A, p.(Ser157Asn) variant in *MYO7A*. Family 5 revealed a 3.7 Mb interval that contained the c.3502C>T, p.(Arg1168Trp) variant in *MYO7A*. The longest interval (13.6 Mb) of the mapping data in family 6 (Table 3) was concordant with the homozygous c.1041dup, p.(Ser348Glnfs*2) variant in *CDC14A* detected by exome analysis. Family 7 revealed a 15.9 Mb interval (Table 3) on chromosome 13, encompassing *SLITRK6* and its homozygous c.120_121insT, p.(Asp41*) variant. The longest interval in family 9 (13.9 Mb, Table 3) included the homozygous *CDH23* c.2968G>A, p.(Asp990Asn) variant. Both intervals in families 10 (3.3 Mb, Table 3) and 11 (4.4 Mb, Table 3) included *GJB2*. Family 12 revealed a 7.7 Mb interval (Table 3) that was consistent with the exome data and includes *MYO15A*. The longest interval in family 13 (29.0 Mb) contained the homozygous *CDH23* c.4688T>C, p.(Leu1563Pro) variant (Table 3).

The most significant linkage intervals in families 3 and 8 did not include the affected genes *GJB2* and *MYO7A*. However, the longest interval in family 3 is located slightly outside of the *GJB2* gene coordinates.

## 4. Discussion

Geographically or culturally isolated populations that have high rates of consanguinity, such as the Pakistani families included in this study, have proven valuable for novel HL gene identification studies and for contributing to a greater understanding of the alleles implicated in HL [46,47,48]. Although our patient cohort was relatively small, the overall resolve rate in this study of 61.9% is nonetheless comparable to other studies with consanguineous families that have showed a resolve rate of 67% [49]. As expected, most of the families revealed variants that were homozygous (92.3%) and compound heterozygous (7.7%) and were primarily found in ARHL-associated genes. 

We identified causative variants in seven HL-associated genes. Many of these variants were missense (53.8%, Figure 3B), which is consistent with the mutational characteristics of deafness genes [50]. Unlike previous studies investigating the genetic spectrum of hearing-impaired Pakistani patients that have described *SLC26A4* as a frequent cause of HL in this population, causal variants in this gene were not present in our cohort. This is likely explained due to the restricted geographical region from which our families were recruited and the existence of prevalent founder variants in this gene, especially in the Pakistani population, that were absent in our relatively small cohort [51].

Genes that are most frequently implicated in autosomal recessive NSHL (ARNSHL) in consanguineous families from Pakistan are *GJB2* (MIM *121011), *SLC26A4* (MIM *605646), *MYO15A* (MIM *602666), *OTOF* (MIM *603681), *CDH23* (MIM *605516), *TMC1* (MIM *606706), *MYO7A* (MIM *276903), and *HGF* (MIM *142409) [6,52,53]. Variants in the genes *MYO7A* and *GJB2* accounted for a combined 46.2% of all diagnoses in the present study which is consistent with previously published rates from Pakistani cohort studies (Figure 3A) [5,6,7]. We also identified the c.231G>A, p.(Trp77*) variant in one family that has been reported as one of the most common *GJB2* alleles in Pakistani HL patients [54].

Interestingly, disease-causing variants in Pakistani patients with NSHL often involved genes that were also associated with syndromic HL, such as *MYO7A* [55] or *CDH23* [39]. For example, patients with a diagnosis of Usher syndrome type 1B (MIM *276900) show profound congenital hearing impairment, retinitis pigmentosa, vestibular dysfunction, and biallelic causal variants in *MYO7A* [56]. In this study, pathogenic variants of *MYO7A* were identified in three families (4, 5, 8). Affected individuals in family 4 and family 5 were confirmed with Usher syndrome, which is characterized by severe auditory and ophthalmic symptoms. Both of the homozygous variants that have been identified in family 4 (c.470G>A, p.(Ser157Asn)) and 5 (c.3502C>T, p.(Arg1168Trp)) are known to cause Usher syndrome type 1 [34,35].

Family 8 revealed three different heterozygous variants in *MYO7A*, with all three of them exclusively present in both affected siblings (IV.1, IV.2), in whom Usher syndrome was excluded. Interestingly, two of the three identified variants (c.1258A>T, c.4505A>G) were previously described in a Pakistani family reporting ARNSHL [6]. Richard et al. [6] also described a third heterozygous variant (c.3502C>T) that differs from the third variant in the present study (c.1849T>C). In family 8, the two c.1258A>T and c.4505A>G variants were inherited from the maternal allele (III.4), and the c.1849T>C variant was an inferred paternally inherited allele, thus confirming compound heterozygosity in both affected patients (IV.1, IV.2) of c.1258A>T, p.(Lys420*) and c.1849T>C, p.(Ser617Pro) (Figure 1). The absent homozygous interval in a region that contains *MYO7A* supports a suspected compound heterozygosity for the variants in this family. It remains to be determined if the double mutated maternal allele is either a broadly segregating allele in the Pakistani population or if the two families are possibly distantly related. 

The homozygous c.2968G>A, p.(Asp990Asn) missense variant in *CDH23* segregating in family 9 was identified as a recurrent variant in South Indian families with HL [57] and is known to cause ARNSHL [39]. The second pathogenic homozygous missense variant in *CDH23* (c.4688T>C, p.(Leu1563Pro)), which is known to cause non-syndromic deafness, has been identified in family 13 [45]. 

We identified two unrelated families (1, 2) with the same c.166C>T, p.(Leu56Phe) variant in the gene *FGF3*. Recessive variants in *FGF3* have been described in patients diagnosed with LAMM syndrome, which is characterized by congenital deafness with labyrinthine aplasia (LA), microtia (M) and microdontia (M) (MIM *610706) [42]. The phenotypic characteristics in patients can vary from fully penetrant LAMM syndrome to milder forms with less severe syndromic features [58]. Probands from families 1 and 2 show HL and cupped ears (Figure 2A), which overlap with milder phenotypic characteristics such as minor dental and external ear phenotypes that were previously described in the literature. We cannot exclude an inner ear malformation in the affected individuals due to absent temporal bone CTs.

Three nonsense variants were found in the genes *GJB2*, *SLITRK6*, and *MYO7A* (Figure 3). While the c.231G>A, p.(Trp77*) variant in *GJB2* and the c.1258A>T, p.(Lys420*) variant in *MYO7A* were previously described as pathogenic, the nonsense variant c.120_121insT, p.(Asp41*) in *SLITRK6*, segregating in family 7, was novel. The effect of the induced stop-codon at amino acid position 41 out of 841 amino acids encoding SLITRK6 would truncate 95% of the protein and likely be targeted by nonsense-mediated mRNA decay (NMD) [59]. To date, only five variants are known to cause the associated autosomal recessive deafness and myopia syndrome [44,60,61] that is consistent with the phenotype in family 7. Myopia in deafness and myopia syndrome has been reported to range between −6 and −11 diopters [44,62]. Findings in both siblings (IV.1, IV.2) in family 7 and affected individuals (IV.3, IV.5) in family 4, who had undergone ophthalmological examination were consistent with high myopia. Additionally, IV.3 and IV.5 in family 4, and IV.2 in family 5 showed findings typical for retinitis pigmentosa. Interestingly, all the previously identified variants are also either nonsense or frameshift variants, suggesting loss-of-function as an underlying mechanism. Consistently, *Slitrk6* knock-out mice showed distinct reduction in cochlear innervation and a defective auditory brainstem response [44].

Family 12 revealed a homozygous splice-site variant c.9518-2A>G in intron 57 of *MYO15A* [5]. This variant likely mediates the complete loss of the 3’ acceptor site according to several in silico prediction tools, and possibly results in the skipping of exon 58. Variants that occur in important consensus sequences at the exon-intron boundaries, thus disrupting the actual splicing process, are known to be the cause of a variety of human diseases [63].

## 5. Conclusions

The present study of 21 Pakistani families identified two novel alleles causing HL and emphasizes the importance of investigating different populations and their heterogeneous genetic background. The fact that 38.1% of the 21 examined families are still considered unresolved highlights a possible area in which the further application of advanced sequencing and analysis methods could uncover currently unrecognized genetic changes due to technical limitations. Candidate genes have been identified in four families that are presently undergoing functional analysis. In families without candidate genes identified, genome sequencing would support uniform copy number variation analysis and the analysis of deep intronic or regulatory variants that are difficult to ascertain by ES. Some of the limitations of the study include potentially insufficient coverage of homopolymeric or GC-rich regions and the existence of mapping difficulties for regions containing pseudogenes or larger deletions, insertions, or structural rearrangements, as listed in previous ES studies, and may diagnose some of the unresolved families [64]. Nevertheless, an ES approach is still the method of choice for elucidating genetic variants in a large portion of heritable human disorders, including HL.

## Figures and Tables

**Figure 1 genes-11-01329-f001:**
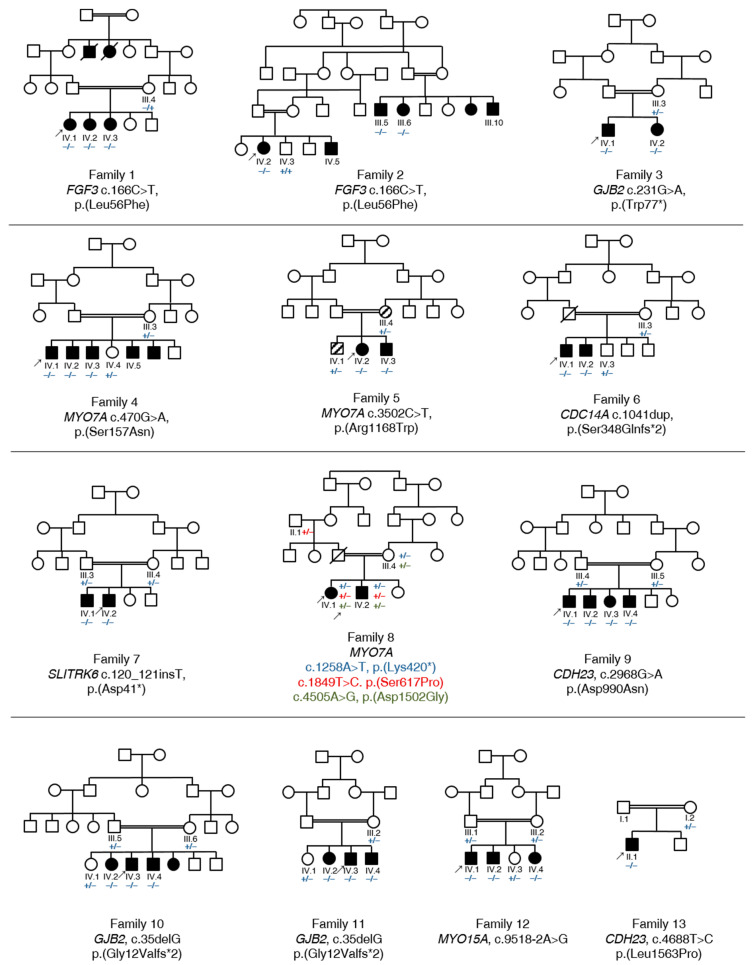
Pedigree and segregation analysis of known and previously undescribed variants in 13 Pakistani families with HL. All the families have a consanguineous background, marked with double lines. Affected individuals are shown in black symbols, and unaffected parents and siblings are shown in unfilled symbols. Individuals with a bone disorder, but without HL, are shown in striped symbols. Probands who were exome sequenced are marked with an arrow. Deceased individuals are marked with a diagonal line. The mutated and wild type alleles are illustrated with “−” (mutated) and “+” (wild type) symbols, respectively.

**Figure 2 genes-11-01329-f002:**
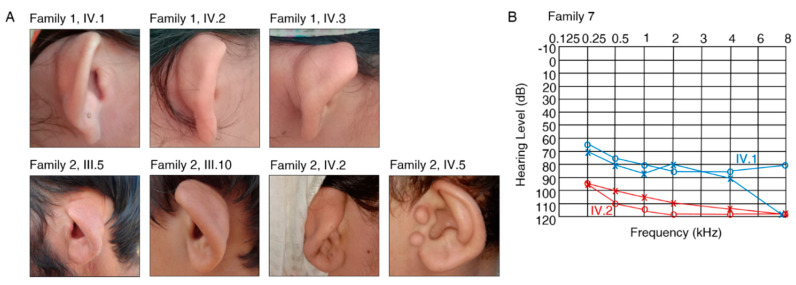
Clinical aspects of patients with previously unreported variants. (**A**) Affected individuals in family 1 (IV.1, IV.2, IV.3) and family 2 (only III.5, III.10, IV.2, IV.5 were available for photographs) show cupped ears and report severe HL. (**B**) Pure-tone audiogram for affected family members IV.1 (blue) and IV.2 (red) in family 7. Left-ear measurements are represented as “x” and right-ear measurements are shown with “o”.

**Figure 3 genes-11-01329-f003:**
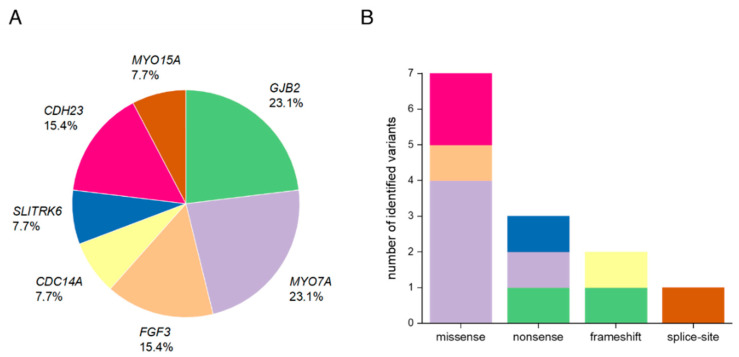
Overview of the affected genes and the distribution of different variant types in Pakistani families with HL. (**A**) Overall percentage of each affected gene in 13 Pakistani families. (**B**) Number of identified variants by type (missense, nonsense, frameshift, splice-site). The color code refers to the genes that are marked in (**A**).

**Table 1 genes-11-01329-t001:** Likely causal variants identified in Pakistani families with hearing loss (HL).

ID	Gene	DFN locus	Transcript	Nucleotide	Protein	Zygosity	MT	PP	SIFT	GERP	LRT	DVD
Family 1	***FGF3***	**--**	**NM_005247.2**	**c.166C>T**	**p.(Leu56Phe)**	1/1	DC	PrD	D	C	U	U
Family 2	***FGF3***	**--**	**NM_005247.2**	**c.166C>T**	**p.(Leu56Phe)**	1/1	DC	PrD	D	C	U	U
Family 3	*GJB2*	DFNB1	NM_004004.5	c.231G>A	p.(Trp77*)	1/1	DC	--	--	C	D	P [33]
Family 4	*MYO7A*	DFNB2	NM_000260.3	c.470G>A	p.(Ser157Asn)	1/1	DC	PrD	D	C	D	P [34]
Family 5	*MYO7A*	DFNB2	NM_000260.3	c.3502C>T	p.(Arg1168Trp)	1/1	DC	PrD	D	C	D	LP [35]
Family 6	*CDC14A*	DFNB32	NM_033312.2	c.1041dup [36]	p.(Ser348Glnfs*2) [36]	1/1	--	--	--	--	--	--
Family 7	***SLITRK6***	--	**NM_032229.2**	**c.120_121insT**	**p.(Asp41*)**	1/1	--	--	--	--	--	--
Family 8	*MYO7A*	DFNB2	NM_000260.3	c.1258A>T	p.(Lys420*)	0/1	DC	--	--	C	D	P [37]
				c.1849T>C	p.(Ser617Pro)	0/1	DC	PrD	D	C	D	U [38]
				c.4505A>G	p.(Asp1502Gly)	0/1	DC	PrD	D	C	D	U [6]
Family 9	*CDH23*	DFNB12	NM_022124.5	c.2968G>A	p.(Asp990Asn)	1/1	DC	PrD	D	C	D	P [39]
Family 10	*GJB2*	DFNB1	NM_004004.5	c.35delG	p.(Gly12Valfs*2)	1/1	--	--	--	--	--	P [40]
Family 11	*GJB2*	DFNB1	NM_004004.5	c.35delG	p.(Gly12Valfs*2)	1/1	--	--	--	--	--	P [40]
Family 12	*MYO15A*	DFNB3	NM_016239.3	c.9518-2A>G		1/1	DC	--	--	C	--	U [5]
Family 13	*CDH23*	DFNB12	NM_022124.5	c.4688T>C	p.(Leu1563Pro)	1/1	DC	PrD	D	C	D	P [41]

1/1 homozygous; 0/1 heterozygous. Previously undescribed variants are marked in bold. Abbreviations: LRT, Likelihood Ratio Test; MT, MutationTaster; PP, PolyPhen-2; SIFT, Sorting Intolerant from Tolerant; GERP, Genomic Evolutionary Rate Profiling; DVD, Deafness Variation Database; C, conserved; D, deleterious; DC, disease causing; P, pathogenic; LP, likely pathogenic; PrD, probably damaging; U, unknown significance.

**Table 2 genes-11-01329-t002:** Clinical information for Pakistani families.

ID	Phenotype	Affected Family Members	Unaffected Family Members
Family 1	HL, cupped ears	IV.1, IV.2, IV.3	III.4
Family 2	HL, cupped ears	III.5, III.6, III.10 ^1^, IV.2, IV.5 ^1^	IV.3
Family 3	HL	IV.1 (25 y/o), IV.2 (10 y/o)	III.3
Family 4	Usher syndrome	IV.1 (35 y/o), IV.2 (33 y/o), IV.3 (32 y/o), IV.5 (33 y/o) ^1^	III.3, IV.4
Family 5	Usher syndrome ^2^, bone disorder ^2^	IV.2 (30 y/o), IV.3 (18 y/o); Usher syndrome	III.4, IV.1; bone disorder
Family 6	severe-to-profound HL, compound myopic astigmatism	IV.1 (30 y/o), IV.2 (28 y/o)	III.3, IV.3
Family 7	severe-to-profound HL, compound myopic astigmatism, glaucoma	IV.1 (26 y/o), IV.2 (23 y/o)	III.3, III.4
Family 8	HL	IV.1 (13 y/o), IV.2 (12 y/o)	II.1, III.4
Family 9	HL	IV.1 (33 y/o), IV.2 (32 y/o), IV.3 (20 y/o), IV.4 (18 y/o)	III.4, III.5
Family 10	HL	IV.2 (14 y/o), IV.3 (13 y/o), IV.4 (12 y/o)	III.5, III.6, IV.1
Family 11	HL	IV.2 (16 y/o), IV.3 (14 y/o), IV.4 (12 y/o)	III.2, IV.1
Family 12	HL	IV.1 (15 y/o), IV.2 (15 y/o), IV.4 (10 y/o)	III.1, III.2, IV.3
Family 13	HL	II.1 (11 y/o)	I.2

Abbreviations: HL, hearing loss; y/o, years old available ages for affected individuals; ^1^ no DNA available for testing, only clinical photographs (Figure 2A) or ophthalmological examination; ^2^ two distinct phenotypes within the family.

**Table 3 genes-11-01329-t003:** Loci for autosomal recessive HL in 13 Pakistani families.

Family ID	Chromosomal Band	Region of Autozygosity Identified by Linkage Analysis (hg19)	Length (Mb)	LOD	Causal Gene in Locus	Gene Coordinates (hg19)
Family 1	11p12-q13.4	39,536,493–73,025,971	33.5	2.529	*FGF3*	chr11:69,624,736–69,634,192
Family 2	11q13.1-q13.3	63,870,810–69,964,525	6.1	3.73	*FGF3*	chr11:69,624,736–69,634,192
Family 3 *	13q12.11-q14.11	22,661,666–41,063,028	18.4	1.927		
Family 4	11q13.3-q14.3	69,063,393–88,489,081	19.4	2.529	*MYO7A*	chr11:76,839,310–76,926,284
Family 5	11q13.5-q14.1	76,792,431–80,457,784	3.7	1.927	*MYO7A*	chr11:76,839,310–76,926,284
Family 6	1p22.2-p21.2	88,430,037–102,069,696	13.6	1.927	*CDC14A*	chr1:100,810,598–100,985,833
Family 7	13q22.1-q31.3	74,995,660–90,925,494	15.9	1.2	*SLITRK6*	chr13:86,366,925–86,373,554
Family 8	No interval close to *MYO7A*					
Family 9	10q21.2-q22.3	64,059,261–78,005,230	13.9	3.006	*CDH23*	chr10:73,156,691–73,575,702
Family 10	13q11-q12.11	19,121,950–22,395,049	3.3	2.529	*GJB2*	chr13:20,761,609–20,767,037
Family 11	13q11-q12.12	19,121,950–23,534,670	4.4	2.529	*GJB2*	chr13:20,761,609–20,767,037
Family 12	17p12-p11.2	13,801,016–21,539,613	7.7	2.529	*MYO15A*	chr17:18,012,020–18,083,116
Family 13	10q21.2-q23.31	61,998,060–91,002,927	29.0	1.2	*CDH23*	chr10:73,156,691–73,575,702

Abbreviations: LOD, logarithm of the odds. * Family 3 revealed a homozygous interval outside but near the *GJB2* gene locus.

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
