# Peer review of "Genetic Spectrum of Syndromic and Non-Syndromic Hearing Loss in Pakistani Families"

_genes, 2020, doi:10.3390/genes11111329_

Round 1

Reviewer 1 Report

The manuscript submitted to GENEs from Julia Doll and coauthors is entitled “Genetic spectrum of syndromic and non-syndromic 2 hearing loss in Pakistani families”

In the abstract the authors state that “21 consanguineous Pakistani families revealed 12 pathogenic and likely pathogenic variants in 39 the genes GJB2, MYO7A, FGF3, CDC14A, SLITRK6, CDH23 and MYO15A with an overall resolve 40 rate of 57.1%.” Overall, the manuscript is nicely crafted, and the table, figures and supplementary figure are of good quality. However, the manuscript offers little that is new beyond what has already been reported, for example, in Richard et al., which Doll and coauthors do cite in their manuscript.

It would be helpful to know more about the families that were not resolved, which might constitute some new and interesting information. Were new deafness loci or deafness genes identified and are going to be reported later? That’s fine. Just say it. If there were no novel deafness loci, also say it. How do the authors explain the apparent hereditary deafness in families in which no mutations were identified that might be causal?  Add some additional thoughts and speculation.

Other issues:

Line 58, Is congenital hearing loss more common than color-blindness?

Line 80, Who among the authors received IRB approval for this study in Pakistan?  How were the probands ascertained initially?  Schools for the deaf?  Audiologist?

If all of the families were ascertained in Pakistan, why only two co-authors from Pakistan and they are middle authors? Who did all the phenotyping, enrollment and blood collection?

Line 91 and 141 have the same information.  Delete sentence about extraction of gDNA from line 91.

Line 140, Provide the primer sequences as a supplementary table.  If a person asks you for these primers a few years form now, the file may be long forgotten/lost.

Line 164,  Add an explanation to the figure legend about the variant for MYO15A which is designated p.(?).  Why is that?

Line 239  Are the two deaf males in Family 6 fertile or infertile?  See Imtiaz et al. 2018 Human Molecular Genetics  https://doi.org/10.1093/hmg/ddx440

Discussion, line 268 Is the "Pakistani population" geographically isolated? Seems to me that there are Pakistani's living all over the world. Are Pakistanis more geographically isolated than, say, Germans? I suggest you find a more sensitive way of making your point. The authors surely never intended this to be a pejorative statement.

Line 297, provide references for nonsyndromic deafness due to variants of CDH23 and MYO7A.

Line 286. Authors have mentioned the most frequent genes associated with deafness in Pakistan. HGF is also a major contributor of deafness in Pakistan. In fact, the 3bp and 10bp non-coding deletions associated with deafness were first identified in a very large number of families from Pakistan (Schultz et.al, 2009).

Line 300, “MYO7A was affected in three families” can it be rephrased to “pathogenic variants of MYO7A were identified in three families”.

Nonsense and frameshift variants of SLITRK6 are associated with deafness and myopia. Was myopia evaluated in proband 1V:2 in family 7?

In Family 12, remove p.(?)   c. designation is suggested for splice site variants.

Were unresolved families from this cohort evaluated for copy number variations. Deletions and duplications can be evaluated using SNP genotyping data.

The authors identified three MYO7A and one CDH23 family in your study (Table 1).  Were those families clinically evaluated for Usher syndrome?

Author Response

Reviewer 1

The manuscript submitted to GENEs from Julia Doll and coauthors is entitled “Genetic spectrum of syndromic and non-syndromic 2 hearing loss in Pakistani families”

In the abstract the authors state that “21 consanguineous Pakistani families revealed 12 pathogenic and likely pathogenic variants in 39 the genes GJB2, MYO7A, FGF3, CDC14A, SLITRK6, CDH23 and MYO15A with an overall resolve 40 rate of 57.1%.” Overall, the manuscript is nicely crafted, and the table, figures and supplementary figure are of good quality. However, the manuscript offers little that is new beyond what has already been reported, for example, in Richard et al., which Doll and coauthors do cite in their manuscript. 

It would be helpful to know more about the families that were not resolved, which might constitute some new and interesting information. Were new deafness loci or deafness genes identified and are going to be reported later? That’s fine. Just say it. If there were no novel deafness loci, also say it. How do the authors explain the apparent hereditary deafness in families in which no mutations were identified that might be causal?  Add some additional thoughts and speculation.

Response: We added the information about newly identified deafness genes in line 395-397.

Other issues:

Line 58, Is congenital hearing loss more common than color-blindness?

Response: Color-blindness is more common than congenital HL. We adjusted the sentence in line 68 (formerly 58).

Line 80, Who among the authors received IRB approval for this study in Pakistan?  How were the probands ascertained initially?  Schools for the deaf?  Audiologist?

Response: Dr. Saad Khan has received the IRB Approval for this genetic study from the Kohat University of Science and Technology in Pakistan, mentioned in section 2.1 (starting line 89). The Pakistani families participating were ascertained through social workers in the remote villages, because they know well who in the regions are affected by hearing impairment.

If all of the families were ascertained in Pakistan, why only two co-authors from Pakistan and they are middle authors? Who did all the phenotyping, enrollment and blood collection?

Response: We have added five more co-authors as we initially planned at the revision stage and changed their order. Enrollment, blood collection and clinical assessment were done locally by Pakistani co-authors and Dr. Hyung-Goo Kim. The positions of some of the original co-authors were changed based on their contribution to this project and the revision phase.

Line 91 and 141 have the same information.  Delete sentence about extraction of gDNA from line 91.

Response: The sentence about the blood extraction was removed from line 101 (formerly 91).

Line 140, Provide the primer sequences as a supplementary table.  If a person asks you for these primers a few years form now, the file may be long forgotten/lost.

Response: We added the primer sequences as a supplementary table (S1) and added the information in line 152 and under “Supplementary Materials” (line 406-408).

Line 164,  Add an explanation to the figure legend about the variant for MYO15A which is designated p.(?).  Why is that?

Response: The designation “p.(?)” was removed from figure 1, table 1 and figure S1 as suggested later in one of the reviewer’s comments. According to HGVS rules, the question mark means the consequence at the protein level is unknown.

Line 239  Are the two deaf males in Family 6 fertile or infertile?  See Imtiaz et al. 2018 Human Molecular Genetics  https://doi.org/10.1093/hmg/ddx440

Response: Unfortunately, we do not have detailed information about the fertility of the two affected individuals, but both of them are unmarried and have no children. This was added in line 272-273.

Discussion, line 268 Is the "Pakistani population" geographically isolated? Seems to me that there are Pakistani's living all over the world. Are Pakistanis more geographically isolated than, say, Germans? I suggest you find a more sensitive way of making your point. The authors surely never intended this to be a pejorative statement.

Response: We re-focused the sentence to only be about our patient cohort in line 307 (formerly 268).

Line 297, provide references for nonsyndromic deafness due to variants of CDH23 and MYO7A.

Response: References for non-syndromic deafness were added for CDH23 and MYO7A in line 337 (formerly 297, highlighted in yellow).

Line 286. Authors have mentioned the most frequent genes associated with deafness in Pakistan. HGF is also a major contributor of deafness in Pakistan. In fact, the 3bp and 10bp non-coding deletions associated with deafness were first identified in a very large number of families from Pakistan (Schultz et.al, 2009).

Response: HGF was added as another frequently involved HL gene with the appropriate reference in line 326 (formerly 286, new reference is highlighted in yellow).

Line 300, “MYO7A was affected in three families” can it be rephrased to “pathogenic variants of MYO7A were identified in three families”.

Response: This sentence was rephrased according to the reviewer’s suggestion in line 340-341 (formerly 300).

Nonsense and frameshift variants of SLITRK6 are associated with deafness and myopia. Was myopia evaluated in proband 1V:2 in family 7?

Response: We have updated the clinical section and added the ophthalmological information for the second affected individual IV.2 in family 7 in line 216-221 (results) and 381 (discussion).

In Family 12, remove p.(?)   c. designation is suggested for splice site variants.

Response: “p.(?)” was removed from figure 1, table 1 and figure S1 for family 12.

Were unresolved families from this cohort evaluated for copy number variations. Deletions and duplications can be evaluated using SNP genotyping data.

Response: CNV analysis was done with the exome data for 19 out of 21 families. We could not identify any deletions or duplications in known HL genes. This information was added in line 148-149. The new reference for the used CNV approach is highlighted in yellow.

The authors identified three MYO7A and one CDH23 family in your study (Table 1).  Were those families clinically evaluated for Usher syndrome?

Response: Family 4 and family 5 that revealed variants in MYO7A were again examined by an ophthalmologist and could be confirmed with Usher syndrome. We added this information to the syndromic HL section (results) in line 196-207 and changed the text accordingly. Family 8 were also examined again and showed normal vision and were not affected by Usher syndrome. This information was added in line 258-259. For family 9 and family 13 that revealed variants in CDH23, no further assessment was possible. Since both variants are known to cause NSHL, we possibly can rule out an Usher syndrome diagnosis.

Reviewer 2 Report

The paper presents the findings from NGS search (exome sequencing, bioinformatics and gene mapping) for causality in 21 consanguineous families. The likely pathogenic variants were identified in 12 families with seven genes involved (success rate 57%). The unsolved 9 families deserve trio exome or trio WGS analysis, which seems to be the plan. Copy number variations and deep intronic and regulatory mutations have not been covered according to the authors’ statement.This kinds of abnormalities might be detected with higher accuracy in WGS.

Audiological data were only presented for probands in family 6 and 7, but they state the the remaining probands all had severe- profound congenital HI. Variants in GJB2 was  a major part (25%), and surprisingly no SLC26A4 variants were identified.

Very limited clinical data are presented, especially regarding the patients with variants in MYO7A and CDH23, known to be associated with dual sensory impairment as Usher syndrome type 1B and type 1D. The ages of the patients are not reported and it is impossible to know if the patients with MYO7A and CDH23 variants have reached ages where retinitis pigmentosa would be expected to cause visual impairment. Apparently no ERGs were done in these patients.The discrimination between syndromic and non-syndromic HI affection is critical for the patients and their future lives.

It is acknowledged that the ascertainment may have been done of families in remote areas of Pakistan with unavailibilty to advanced medical service, but this circumstance of limited details clinically weakens the report, and the transfer value for future patients with identical mutations.

Segregation analysis of identified variants have been performed in most families.

The paper is well written with clear instructive tables and figures, and very detailed.

Despite these elements the paper does not add critical new information about genetics in the Pakistani population which has been characterized in numerous previous publications.

Author Response

Reviewer 2

The paper presents the findings from NGS search (exome sequencing, bioinformatics and gene mapping) for causality in 21 consanguineous families. The likely pathogenic variants were identified in 12 families with seven genes involved (success rate 57%). The unsolved 9 families deserve trio exome or trio WGS analysis, which seems to be the plan. Copy number variations and deep intronic and regulatory mutations have not been covered according to the authors’ statement.This kinds of abnormalities might be detected with higher accuracy in WGS.

Response: As correctly assumed, we plan to verify some novel candidate genes we identified in the remaining families. In some families with no candidate genes, we will perform WGS to detect cryptic mutations in introns and regulatory elements. Also, WGS will be able to detect CNV, which might be involved in some families. We clarified this in the final paragraph of the discussion (lines 393-403).

Audiological data were only presented for probands in family 6 and 7, but they state the the remaining probands all had severe- profound congenital HI. Variants in GJB2 was a major part (25%), and surprisingly no SLC26A4 variants were identified.

Response: The affected members of all families had congenital hearing loss. Importantly, mothers raised their concern about hearing in their infants and it was confirmed at the later stage as a profound hearing loss. Since most of them could not afford to buy hearing aids, their visit to ENT doctors has not transpired. But their severe symptoms have been confirmed by our Pakistani coauthors listed on the manuscript.

Very limited clinical data are presented, especially regarding the patients with variants in MYO7A and CDH23, known to be associated with dual sensory impairment as Usher syndrome type 1B and type 1D. The ages of the patients are not reported and it is impossible to know if the patients with MYO7A and CDH23 variants have reached ages where retinitis pigmentosa would be expected to cause visual impairment. Apparently, no ERGs were done in these patients. The discrimination between syndromic and non-syndromic HI affection is critical for the patients and their future lives.

It is acknowledged that the ascertainment may have been done of families in remote areas of Pakistan with unavailibilty to advanced medical service, but this circumstance of limited details clinically weakens the report, and the transfer value for future patients with identical mutations.

Response: We added the ages of affected patients with variants in CDH23 and MYO7A (family 4, 5, 8, 9, 13) (plus all affected individuals that were available for this information) in table 2 and assessed three families with variants in MYO7A by an ophthalmologist. We could confirm Usher syndrome in two families (family 4 and 5) (lines 196-207).

Segregation analysis of identified variants have been performed in most families.

The paper is well written with clear instructive tables and figures, and very detailed.

Despite these elements the paper does not add critical new information about genetics in the Pakistani population which has been characterized in numerous previous publications.

Response: We would like to highlight that our manuscript has information about the prevalence of some known genes involved in autosomal recessive hearing loss/impairment in Pakistani consanguineous families from the Peshawar region. Our cohort with 13 diagnosed families includes three families each with different MYO7A and GJB2. Two families had FGF3 variants. Given the critical new information that will come from the announcement of novel autosomal recessive genes, our work successfully excludes some families with variants in known genes for subsequent genetic analysis of other families with novel disease genes.

Reviewer 3 Report

it would be better to expand the clinical evaluation item because it seems to be short. Only families 6 and 7 had ophthalmic evaluation ? Only families 6 and 7 had audiologic evaluation ? any sex related hearing loss? all congenital? Ages at diagnosis?

In limitations: to study a special ethnic population with high cosanguinity reduces the chance to find a wide spectrum of genetic variations and near each day appears a new genetic cause of hearing loss, there is a lot of work to be done in this field to cover all populations and genetic hearing loss.

Author Response

Reviewer 3

it would be better to expand the clinical evaluation item because it seems to be short. Only families 6 and 7 had ophthalmic evaluation ? Only families 6 and 7 had audiologic evaluation ? any sex related hearing loss? all congenital? Ages at diagnosis?

Response: The hearing loss found in our Pakistani cohort is all congenital and thus the diagnosis was done within one year of the age of infant. Ophthalmic evaluation has been performed only in families 4, 5, 6, 7 and 8. We have clinically clarified Usher syndrome in two of the families with MYO7A variants. Among our Pakistani cohort, there is no family with X-linked inheritance.

In limitations: to study a special ethnic population with high cosanguinity reduces the chance to find a wide spectrum of genetic variations and near each day appears a new genetic cause of hearing loss, there is a lot of work to be done in this field to cover all populations and genetic hearing loss.

Response: In the literature, there have been many autosomal recessive disease genes identified in one ethnic population. In subsequent studies after the identification of genes, different variants in the same genes have later on been found in other ethnic populations with consanguineous marriages. This has been seen in Iranian, Pakistani, Saudi Arabian, and Turkish patients. In Western countries, where consanguineous marriages are uncommon, the compound heterozygous variants inherited from each parent have been found in the same genes. Studying a specific ethnic population with high consanguinity INCREASES the chance to find a wide spectrum of genetic genetic variations in other ethnic populations.

Round 2

Reviewer 2 Report

Genetic spectrum of syndromic and non-syndromic hearing loss in Pakistani families By Doll J et al

The authors have revised the manuscript and added some clinical data as well as ages of the individuals in a number of cases.

By way of ophthalmological examination (no details whether ophthalmocospy was supplied with ERG, nor any degree of subjective symptoms are given) it is stated that the MYO7A in fam 4 and 5 was associated with Usher syndrome. The patients in both families were adult so it would be expected to know about restricted visual fields and impaired night vision?

In fam 8, Usher was excluded, but no details about which information/examinations were behind that statement.

Regarding CDH23 mutations in fam 9 and 13, all patients were adults but no information about possible subjective affection.The authors indicate that the patients had non-syndromic HI rather than Usher and merely base it upon two publications reporting the same mutations (ref number 39 and 41), but since there are cases with discrepant clinical affection (Usher or only isolated deafness) in some CDH23 mutations in different branches with exactly the same mutation this comparison with published isolated HI associated with the identified CDH23 mutations can only partly exclude retinal affection. Examination of the subjects in this manuscript should have a detailed eye assessment.

The authors explain intentions of future studies to better cover the detection of CNV’s and identification of new autosomal recessive genes, which unfortunately does not count in terms of increasing the novelness of genetic information in the present paper, which mainly adds to several previous genetic epidemiological publications on HI/deafness in Pakistan, rather than provide critical new data..

Author Response

Genetic spectrum of syndromic and non-syndromic hearing loss in Pakistani families by Doll J et al

The authors have revised the manuscript and added some clinical data as well as ages of the individuals in a number of cases.

By way of ophthalmological examination (no details whether ophthalmocospy was supplied with ERG, nor any degree of subjective symptoms are given) it is stated that the MYO7A in fam 4 and 5 was associated with Usher syndrome. The patients in both families were adult so it would be expected to know about restricted visual fields and impaired night vision?

Response: We added detailed information about the ophthalmological examination and symptoms for the Usher families 4 and 5 on lines 198-204 (family 4, IV.3 and IV.5) and 209-212 (family 5, IV.2). We included another sentence about retinitis pigmentosa in the discussion section in line 395-396.

In fam 8, Usher was excluded, but no details about which information/examinations were behind that statement.

Response: We included the requested information about the ophthalmological examination and the exclusion of Usher syndrome in family 8 on lines 268-271.

Regarding CDH23 mutations in fam 9 and 13, all patients were adults but no information about possible subjective affection. The authors indicate that the patients had non-syndromic HI rather than Usher and merely base it upon two publications reporting the same mutations (ref number 39 and 41), but since there are cases with discrepant clinical affection (Usher or only isolated deafness) in some CDH23 mutations in different branches with exactly the same mutation this comparison with published isolated HI associated with the identified CDH23 mutations can only partly exclude retinal affection. Examination of the subjects in this manuscript should have a detailed eye assessment.

Response: Unfortunately, family 9 and family 13 were not readily available for further ophthalmological assessment but they reported that all affected family members (HL) are not affected by additional visual impairment. Both CDH23 variants c.2968G>A, p.(Asp990Asn) and c.4688T>C, p.(Leu1563Pro) were clearly associated with an isolated form of non-syndromic HL (Bork et al. 2001 (Nr. 39), Schultz et al. 2011 (Nr. 41) and were curated as pathogenic by the Deafness Variation Database (DVD) and HGMD and likely pathogenic by ClinVar for causing ARNSHL. They described families that were either affected by ARNSHL or Usher syndrome and could clearly distinguish between individual variants in CDH23 and the association with the HL phenotype. We agree that families with variants in the gene CDH23 should undergo further ophthalmological examination to be sure that visual impairment was not overlooked, but unfortunately, this was not possible in these two families.

The authors explain intentions of future studies to better cover the detection of CNV’s and identification of new autosomal recessive genes, which unfortunately does not count in terms of increasing the novelness of genetic information in the present paper, which mainly adds to several previous genetic epidemiological publications on HI/deafness in Pakistan, rather than provide critical new data..

Response: The focus of this study was to describe our findings involving pathogenic/likely pathogenic variants in this cohort. Many of these families have been resolved and others require further analysis and functional studies. We originally refrained from discussing candidate genes in the first submission. However, Reviewer 1 in the previous revision comments requested that we simply state if we have identified candidate genes or not. As our strategic plans outline that we do not discuss these genes without having first performed some level of functional analyses (they presently do not meet the ACMG “likely pathogenic” classification), we refrain from describing these for now.